# Fibre Optic FBG Sensors for Monitoring of the Temperature of the Building Envelope

**DOI:** 10.3390/ma14051207

**Published:** 2021-03-04

**Authors:** Janusz Juraszek, Patrycja Antonik-Popiołek

**Affiliations:** Faculty of Materials, Civil and Environmental Engineering, University of Bielsko-Biala, 43-309 Bielsko-Biala, Poland; pantonik@ath.bielsko.pl

**Keywords:** monitoring, wall temperature, fibre bragg grating sensors

## Abstract

Standard sensors for the measurement and monitoring of temperature in civil structures are liable to mechanical damage and electromagnetic interference. A system of purpose-designed fibre optic FBG sensors offers a more suitable and reliable solution—the sensors can be directly integrated with the load-bearing structure during construction, it is possible to create a network of fibre optic sensors to ensure not only temperature measurements but also measurements of strain and of the moisture content in the building envelope. The paper describes the results of temperature measurements of a building 2-layer wall using optical fibre Bragg grating (FBG) sensors and of a three-layer wall using equivalent classical temperature sensors. The testing results can be transmitted remotely. In the first stage, the sensors were tested in a climatic test chamber to determine their characteristics. The paper describes test results of temperature measurements carried out in the winter season for two multilayer external walls of a building in relation to the environmental conditions recorded at that time, i.e., outdoor temperature, relative humidity, and wind speed. Cases are considered with the biggest difference in the level of the relative humidity of air recorded in the observation period. It is found that there is greater convergence between the theoretical and the real temperature distribution in the wall for high levels (~84%) of the outdoor air relative humidity, whereas at the humidity level of ~49%, the difference between theoretical and real temperature histories is substantial and totals up to 20%. A correction factor is proposed for the theoretical temperature distribution.

## 1. Introduction

Practical temperature and humidity calculations are based on the values of outdoor and indoor temperatures and characteristics of the building envelope materials. Apart from temperature, however, they usually fail to take account of the variability of more weather, environmental and other factors.

The classical method of analysing the temperature distribution in the building envelope consists of embedding temperature sensors (typically electrical resistance sensors) in it during the building construction or renovation. This solution was applied during the renovation of a three-layer building envelope by placing sensors in four layers [1] The testing focused on the temperature distribution and the possibility of the occurrence of water vapour condensation. The applied three-layer system was composed of a brickwork wall, polyurethane insulation with an aluminium foil layer, an air gap, and a brickwork wall, which worked well. Another way is to scan the building envelope surface using a thermographic camera. The methodology of the testing is discussed in detail in [2]. The study also presents the method of a theoretical calculation of temperature on the outer surface of a multilayer building envelope exposed to 24-h changes in ambient temperature caused by solar radiation and measured using a thermographic camera.

The FBG sensors are the latest solution to use for temperature measurements. The use of fibre optic sensors to measure temperatures in tight walls to monitor potential underground leakages of fluids is discussed in [3,4,5,6]. The test results made it possible to state that the temperature of the fibre optic sensors varied with a change in the seepage rate. The results indicate unequivocally that the use of optic fibre sensors in the building envelope is one of the best solutions that will enable the most accurate analysis of the temperature distribution and of the way the distribution is affected by other external factors.

An interesting series of thermal testing of glued laminated timber beams used in the construction industry performed with a measuring system based on the fibre Bragg grating (FBG) principle was carried out in a thermal chamber [7]. The chamber can generate specific cycles of temperature and humidity to simulate the structural behaviour in real environmental conditions. The tested FBG system is suitable for load-carrying timber (glued laminated timber beams) and concrete structures [8]. The second part of the paper presents the testing of mechanical loads of a glued and laminated timber beam with embedded FBG sensors.

The fibre Bragg grating is an element in a glass or polymer fibre with periodic modulation of the refractive index along with the fibre core. The fundamentals of the FBG-based method of strain and temperature measurements are presented in [9,10].

Fibre optic sensors have many advantages, such as high accuracy, ease of installation, small size, flexibility, cost-effectiveness, resistance to electromagnetic interference and corrosion, multiplexing capability, and great potential in long-term continuous measurements [11,12,13,14,15,16].

An intelligent system of monitoring a concrete paving slab subjected to the impact of the outdoor temperature is presented in [17]. Truly distributed fibre optic temperature and strain sensors and an innovative interferometer-based fibre optic inclinometer were installed in the paving slab to monitor and evaluate the thermal curing process.

The results confirm that the paving slab top surface responds to heating/cooling very fast. The temperature of the bottom surface lags behind the temperature of the top, which results in a rapid rise and then a slow reduction in the temperature difference between the top and the bottom.

A large number of quasi-distributed FBG sensors were presented in [18,19,20]. The aim of the testing was to develop and demonstrate an integrated comprehensive system for the monitoring of thermal shrinkage of the roadway surfacing. The technology has been used for measuring the temperature, strain, and deflection of surfacing since 2003.

It should be noted that the fibre Bragg grating is a promising measuring technology for future applications of sensor systems. The FBG sensor housing should be compatible with the housing of the electrical sensor so that it can be installed easily in systems of instrumentation [21].

Other implementations of FBG sensors made by the author included their application for strain analysis of power transmission line towers, strain analysis of the hoisting machine brake lever, and strain monitoring of a residential building made in the polystyrene concrete technology [22,23]. Interesting study on the temperature distribution of the earth’s surface was presented in the article [24].

Summing up the results of the literature survey, it should be emphasised that conventional temperature sensors have certain limitations, such as poor operational life, low resolution, sensitivity to environmental factors-strong alkalinity, and water permeability (building walls), in particular, low zero stability and high measurement noise. FBG sensors are an alternative method of monitoring building structures due to a number of significant advantages, e.g., high accuracy and sensitivity, flexibility, resistance to electromagnetic interference, and harsh environmental conditions; they are also multiplexable (a large number of sensors can be connected to one telecommunication fibre optic cable, which means simpler cabling systems). A very interesting feature is the possibility of measuring different physical quantities, for example, temperature and strain, using one optical interrogator. The FBG technique is more expensive compared to classical methods of temperature measurement, but considering the technological progress in the field of interrogators and sensors, especially with the wavelength of 800 nm, it will become comparable in terms of price offering much greater research possibilities at the same time.

The approach based on special optical FBG temperature sensors to monitor the temperature distribution in a building wall has never been proposed before. Another novelty is the application of optical FBG sensors to measure not only the wall temperature but also the wall strain. The proposed FBG sensors have a very small diameter, which is of special significance because they can be installed in existing walls of already operated buildings. Temperature measurements in external walls of buildings are necessary because based on them, it is possible to select the optimal heating system depending on the wall structure and specific climatic conditions of the building location. Moreover, the knowledge of real delays of the temperature distribution in the wall due to changes in the outdoor temperature makes it possible to take adequate action in the HVAC system in advance. Furthermore, the knowledge of the real temperature distribution in the wall enables verification of the theoretical (standard) temperature distribution, the introduction of a correction factor, better selection of the heating system parameters, and more precise identification of places in the wall where water vapour condensation occurs. The rationale for presenting walls with traditional and optical FBG temperature sensors in the paper was the need to draw attention to a rather high failure frequency of the former. What is also important, previous studies do not take up the issues of the life and failure frequency of classical sensors intended for measurements of temperature. These measurements are carried out on a long-term basis (over a period of many years). A failure of a temperature sensor and the time drift create serious problems in the analysis of results. The operational analysis of classical temperature sensors embedded in the building walls conducted by the authors over the period of four years showed their 19% failure rate. In a building that is already operated their replacement is practically impossible, which is a significant problem. The number of different sensors installed in the analysed facility was about 3000. The only feasible option in an already operated building is to introduce optical FBG sensors into the wall in place of the damaged classical ones. It should also be noted that in the same period of observation no failure occurred of any of the optical FBG sensors. FBG sensors can be installed in place of damaged traditional sensors due to their small dimensions, and that was conducted also in the case of the analysed three-layer wall, which may not have been described precisely enough. FBG sensors are equivalent to classical sensors and they also demonstrate a number of advantages that the latter are practically unable to achieve. The system of temperature measurements in the building wall presented in the paper and based on optical FBG sensors ensures unprecedented measuring stability, operational durability, and the possibility of creating networks. Moreover, it enables strain and humidity measurements. The significant development of the fibre optic technology, and especially of the design of optical interrogators and fibre optic sensors, will make it possible to introduce into the building structure a “nervous system” that comprehensively and simultaneously monitors the temperature, strain, stress, or vibration of the building walls. Considering the latest technological progress in the field of optical interrogators and optical FBG sensors, their cost is comparable to classical methods of temperature measurement in the building wall, whereas the computational cost is acceptable for large facilities. The outdoor temperature in both cases of the walls under analysis was obtained from a weather station, while the indoor temperature was measured with a sensor type Testo 605i. The theoretical distribution of temperature is determined in the paper-based analytical formulas and is included in the EN ISO 13788:2013-05 standard.

Striving to reduce energy consumption is increasingly associated with the introduction of “smart” technologies into civil engineering structures and the application of systems monitoring and controlling the processes occurring in buildings. The paper presents a proposal for purpose-designed FBG sensors intended for the measurement of temperature in the building envelope. The analysis concerns a three-layer building envelope.

A thermal analysis was conducted of individual layers of the building envelope depending on varying outdoor weather conditions. From the point of view of the building envelope heating and cooling, it is essential what external factors affect the temperature in it and how long it takes the envelope to respond to changes in the outdoor temperature. FBG sensors were used, which—as previously stated—are a very promising measuring technology for future applications in systems of sensors to measure temperatures, strain, displacement, and even moisture content. Experimentally, through-bore sets of FBG sensors were made to measure temperature. The sensors were fixed in a special housing.

Strain measurement of the building structural elements by means of FBG sensors makes it possible to detect the formation of cracks in the building structure, strains arising during the foundation settlement, analysis of volumetric shrinkage of the concrete floor and reinforced concrete columns, or analysis of strains due to standard loads or load tests. Because research in this field falls into a different category, the results will be presented in a separate paper. The paper includes example measurement results of the wall strain recorded using FBG sensors.

## 2. Materials and Methods and Preliminary Testing

The tests were performed on two selected walls—an external cavity wall (three-layer wall) and an external insulated solid wall (two-layer wall) of a university building. The cross section of the walls is presented in Figure 1 and Figure 2, respectively (λ-coefficient of thermal conductivity and thickness of the layers).

The first wall consists of hollow clay blocks, a thermal insulation layer, and clinker bricks. The second—of hollow clay blocks and thermal insulation finished with plaster. These wall types are the most common solutions for the two- and three-layer systems. The first wall was analysed using classical temperature sensors, whereas the second was tested with FBG sensors. The environmental conditions in the terms of air temperature, wind speed, and air humidity were monitored by a classical meteorological station located on the university campus. A temperature monitoring system was constructed inside the building envelope walls. The system is based on fibre Bragg grating (FBG) temperature sensors. The essence of the solution is to introduce FBG sensors at the appropriate points of the cross section of the wall of the structure. In the case of the building envelope in the form of a wall, the appropriate places are the borders of individual layers and points in the middle of a given layer. The places were pre-determined using numerical simulations.

The FBG-based set of temperature sensors is presented in Figure 3. Based on preliminary testing, the total measuring base was adopted as L = 450 mm. The system of FBG sensors was fixed in the building envelope using special holders.

The measuring system calibration is presented in Table 1.

The measuring part of each sensor is a Bragg grating characterised by a specific wavelength and embedded in an optical fibre. Due to the sensor identifiability by the optical interrogator, the difference of at least 5 nm has to be kept between the wavelengths of each Bragg grating. A temperature change in a selected point of the building envelope is closely related to the change in the Bragg grating wavelength. This relationship is described by the following Equation (1):(1)$$T=T_{S1}{\left(\frac{\lambda_{T,act}-\lambda_{T,ref}}{\lambda_{T,ref}}\right)}^{2}+T_{S2}\left(\frac{\lambda_{T,act}-\lambda_{T,ref}}{\lambda_{T,ref}}\right)+T_{S3}$$

The description of constant calibration parameters is presented in Table 2, whereas the values of the calibration constants for individual sensors are listed in Table 3.

Apart from dedicated fibre optic sensors, the system includes a 2 kHz FBG-800 optical interrogator, a recorder, special software, a multiplier and telecommunication fibres. An option is also possible with a wireless transfer of measurement results from the interrogator. The sampling frequency in the case of the temperature measurement in the building envelope was set at the level of 1 Hz, which gives 172,000 measurements per day. A special program was developed to sample the results every 1 min.

The constructed fibre optic system intended for the measurement of temperature inside the external wall was first tested in laboratory conditions. The tests consisted of setting known values of temperature in the climatic chamber containing the set of sensors and presented in Figure 4. The tests were carried out in the Laboratory of Geosynthetics of the University of Bielsko-Biala, Poland. The laboratory is equipped with a climatic chamber with a certified temperature measurement system.

The temperature inside the chamber was additionally monitored using a temperature sensor with an accuracy of 0.1 °C. The essence of the calibration calculations was to record first the heating and then the cooling process in the range of temperatures from −20 °C to 40 °C. The process was repeated to determine the measurement uncertainty and hysteresis. The testing resulted in characteristics illustrating wavelength variations for every sensor. The characteristics are presented in Figure 5. The curves illustrate changes in the FBG sensor wavelength depending on temperature changes in the climatic chamber.

The obtained results confirm the findings of other studies, where a change by 1 degree caused a change in the wavelength by 0.007–0.01 nm [7].

## 3. Results

During the testing, a high-resolution thermographic camera (type: Testo 872, Testo Sp. Z o.o., Warsaw, Poland) was also used to validate the temperature inside and outside the building. The differences in temperature between the surface of the building envelope and ambient air are determined with a thermographic camera from a single thermogram presenting the image of the envelope surface and of the object receiving the air temperature (cf. Figure 6). The object receiving the air temperature should be characterised by low thermal capacity (the change in its temperature will follow the variations in the air temperature) and a matt surface with a high emissivity value. The object should be placed about 20–30 cm away from the surface of the imaged wall. A folded sheet of matt paper can be used for example. The temperature differences specified thermographically should be determined as differences in mean temperature values in a certain field and not in a measuring point. This approach improves accuracy considerably. Because the value of the temperature difference is derived from the thermogram, quantitative thermographic measurements of buildings should be carried out using high-accuracy cameras. It is also significant that the method of thermal imaging is absolutely non-invasive (it does not disturb the investigated temperature field and has no destructive impact on the object whatsoever), and it is possible to perform remote measurements, which is of special importance in the case of testing historical building or buildings where direct access to the surface is difficult. The biggest advantages of thermography are the possibility of making fast measurements and the visual form of the result—the thermogram. The air temperature and humidity were monitored inside the room using the Testo 605i probe. The results of the external wall temperature measurements performed using a thermographic camera are presented in Figure 6. The lowest recorded value was −7.1 °C.

Table 4 presents the values of temperature of the external wall of the analysed building envelope—points M1 and M2.

The results of the measurements of the wall temperature from the inside (the laboratory room) are presented in Figure 7. To determine the air temperature in the room adjoining the analysed building envelope, an additional measurement was performed using the camera with a white sheet of paper placed 40 cm away from the wall in the bottom left-hand corner. The measured temperature of the surface of the paper sheet, i.e., of the room air in this point totalled 18.4 °C, whereas the wall temperature was included in the range of 17.5–18.4 °C (cf. Figure 7).

### 3.1. Two-Layer Wall

The testing of the temperature distribution using the FBG sensors was started in the two-layer wall. Two fibre optic sensors were placed on the border of the layers. Depending on the needs, the sensors could be moved along the building envelope thickness. This technological solution makes it possible to reduce the number of FBG sensors to the minimum. Moreover, the temperature sensors can be used many times. The space between the inside of the building envelope and the guide of the unit with the FBG sensors was filled with a special gel. This substantially reduces the costs of the FBG temperature monitoring due to the possibility of the multiple uses of the sensor unit. FBG sensors are equivalent to classical temperature measurement sensors used in the three-layer wall.

Considering the large number of data obtained for the two-layer wall, a 24-h measurement was performed. The outdoor temperature in the analysed periods varied in the range from −1 °C to 4 °C. Figure 8 illustrates the temperature distribution determined based on previously performed calibration tests in the calibration chamber.

The temperature distribution determined experimentally inside the building envelope based on the measurements performed using fibre optic sensors and a thermographic camera is presented in Figure 9.

### 3.2. Three-Layer Wall

The temperature distribution was also analysed for the three-layer wall. At the stage of the building construction, classical temperature sensors were introduced with relevant infrastructure. During the building operation, sensor II was damaged and replaced with an equivalent FBG sensor. The environmental data related to the air outdoor temperature and relative humidity were obtained from a weather station. The process of temperature changes in the building envelope was analysed for extreme temperature conditions. Compared to positive values of temperature, the temperature in the building envelope was much more stable in the case of negative temperatures. Analysing the distributions determined experimentally, it can be noticed that the temperature inside the envelope does not fall below −8 °C even if the outdoor temperature keeps at the level of −14 °C for more than 5 h. It is only after the outdoor temperature rose to the temperature level in the building envelope that a slight increase in temperature could be observed between the external layer (1) and the thermal insulation. From the moment when the outdoor temperature rises at 8:15, 3 h pass before the temperature inside the building envelope starts to increase (cf. Figure 10 and Figure 11).

In the case of positive values of the outdoor air temperature, the temperature in the building envelope is less stable and rises due to heat accumulation to levels higher than the outdoor temperature (cf. Figure 12). At the maximum outdoor temperature of 35 °C, the temperature in the air gap reaches over 38 °C. The delay in the temperature change in layer 2 due to the temperature drop from 16:45 h onwards totals 3 h. The delay totals about 3 h both for negative and positive values of outdoor temperatures.

The performed analyses also drew attention to differences between the theoretical and the real temperature distribution. They become visible especially if the air relative humidity is included in the range of 40–50% (cf. Figure 13). The theoretical and real temperature distributions in the building envelope obtained for the ambient air relative humidity of 46% and outdoor temperature of 6 °C are presented in Figure 13. Figure 14 presents the distributions for a relative humidity of 91% and outdoor temperature of 9 °C. For high values of relative humidity (90–100%), a good agreement can be observed between the theoretical and experimental temperature distribution (agreement up to 1 °C). The difference between the values determined experimentally (14.57 °C) and theoretically (18.06 °C) for the case with relative humidity at the level of 46% and the air outdoor temperature of 6 °C totals 3.49 °C. It is an essential difference, which reappeared in subsequent measurements.

As already mentioned, optical FBG sensors also make it possible to determine strain values. Example strain measurement results obtained for a wall under a 5 kN load from the floor slab placed above it are presented in Figure 15. A relatively small strain of about 7 μstrain (a millionth) can be observed together with n measurements performed by the FBG-800 optical interrogator with a measuring frequency of 100 Hz. This type of sensor enables structural monitoring of building structures.

## 4. Discussion

Correlation analysis (linear correlation) was conducted to establish the relationship between the air outdoor temperature and the temperature in the three-layer envelope of the building. The correlation is very strong (>75%) and the results are listed in Table 5 with respect to individual sensors embedded in the building envelope. The correlation strength decreases with the analysis of subsequent sensors located closer to the inside of the room.

Based on the temperature differences inside the analysed building envelope between the values determined experimentally and theoretically, a correction function was developed for temperature distributions with the air relative humidity higher than 60% in the form of a fifth-degree polynomial. The polynomial is expressed as follows:(2)$$\Delta t=\;y\;=\;83.772x^{4}\;-\;48.362x^{3}\;-\;50.793x^{2}\;+\;29.504x\;+\;0.4736$$
where x is the total thickness of the building envelope from the internal wall to the analysed point of temperature determination (usually the border between the layers).

To correct the theoretical temperature in a given point of the building envelope, a correction factor has to be added as follows:(3)$$t_{x}=t_{t}\;+\;\Delta t_{L}$$
where

*t_x_*—corrected temperature;

*t_t_*—theoretical temperature;

$\Delta t_{L}$—correction factor for the theoretical temperature distribution at a low humidity value.

Figure 16 presents the implementation of the polynomial approximation of the temperature differences of the envelope.

Table 6 presents polynomial approximation for an example of the distribution of the theoretical temperature in the analysed envelope.

The developed approximation function applied to other temperature distributions makes it possible to correct the theoretical distribution and bring it closer to the real one. Figure 17 and Figure 18 present the theoretical and the real temperature distribution together with the distribution corrected using the developed correction factor for two selected temperature measurements at the air humidity of 46% and 24% and the outdoor temperature of 9 °C and 8 °C, respectively.

## 5. Conclusions

The paper presents an innovative method of monitoring the temperature of the building envelope using FBG temperature sensors. The following conclusions can be drawn:It is demonstrated that the fibre optic technology based on multiplexed FBG sensors enables effective measurement of temperature in the building envelope. The thermal boundary conditions on the outside of the analysed envelope can be obtained by means of measurements performed using a thermographic camera. FBG sensors can be used instead of classical temperature sensors, especially if the latter types are damaged. The hybrid method of the measurement of temperature distributions consists of implementing various measuring methods, which enhances the effectiveness of performed measurements and may contribute to a reduction in the costs of conducting the experiment;The calibration tests demonstrate a linear dependence between temperature and the wavelength measured using fibre optic sensors;Differences are observed between the theoretical and the real temperature distribution in the building envelope for the ambient air relative humidity level of 40–50%;A correction factor is proposed to correct the theoretical temperature distribution in the building envelope. The factor makes it possible to eliminate the difference between the theoretical and the real temperature distribution;Considering the latest technological progress in the field of optical interrogators, the cost is comparable to classical methods of temperature measurements in the building wall. In fact, it can even be included in the low-cost group, while the computational cost is acceptable for large facilities;In the future, the system presented in the paper will enable comprehensive thermal and structural monitoring of building structures. The determined real thermal delays occurring in the wall will make it possible to optimise the building heating/cooling strategy. The knowledge of temperature distributions in the building walls depending on climatic conditions will enable the selection of a heating system suitable for a given wall configuration, which may reduce energy consumption.

## Figures and Tables

**Figure 1 materials-14-01207-f001:**
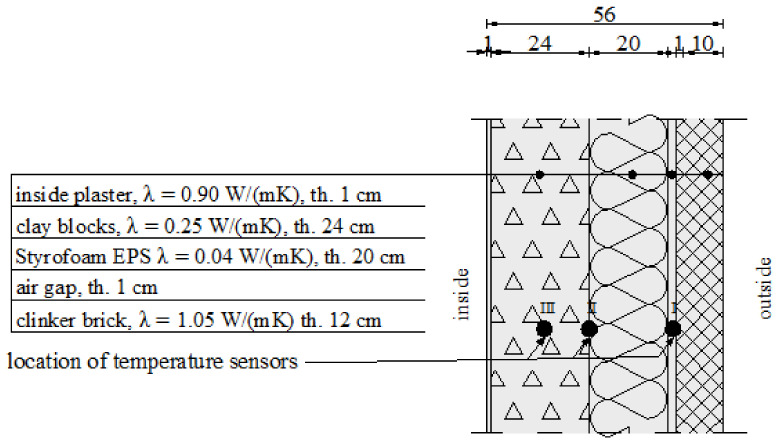
Three-layer wall cross section.

**Figure 2 materials-14-01207-f002:**
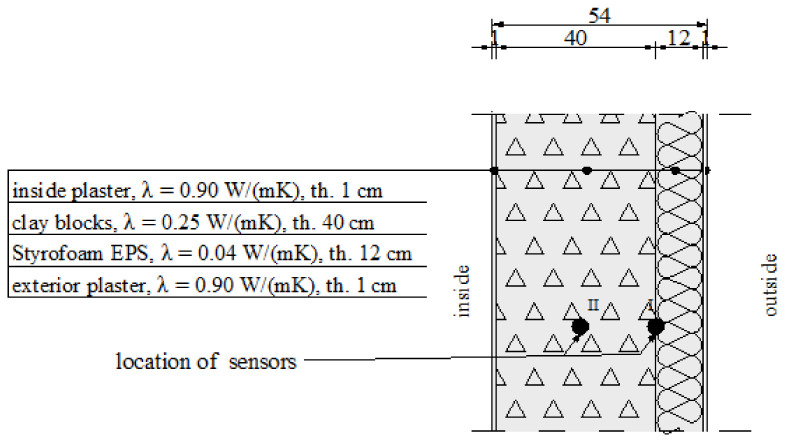
Two-layer wall cross section.

**Figure 3 materials-14-01207-f003:**
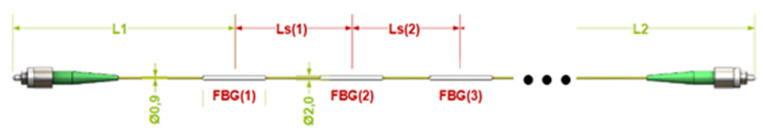
Fibre Bragg grating (FBG) temperature sensors.

**Figure 4 materials-14-01207-f004:**
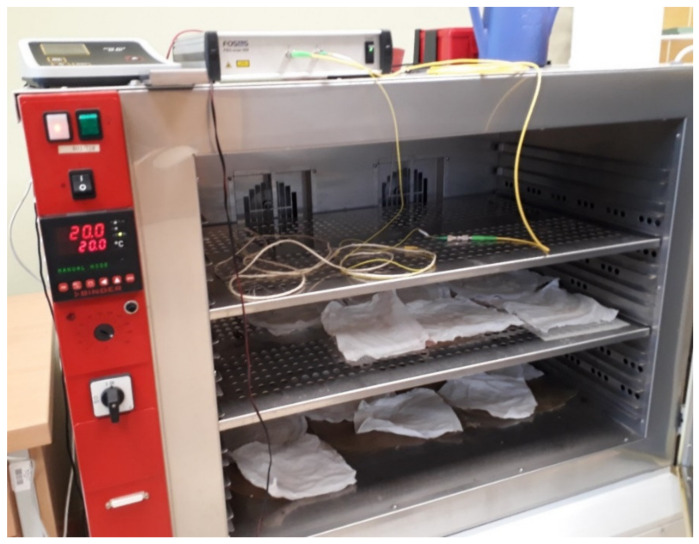
Climatic chamber.

**Figure 5 materials-14-01207-f005:**
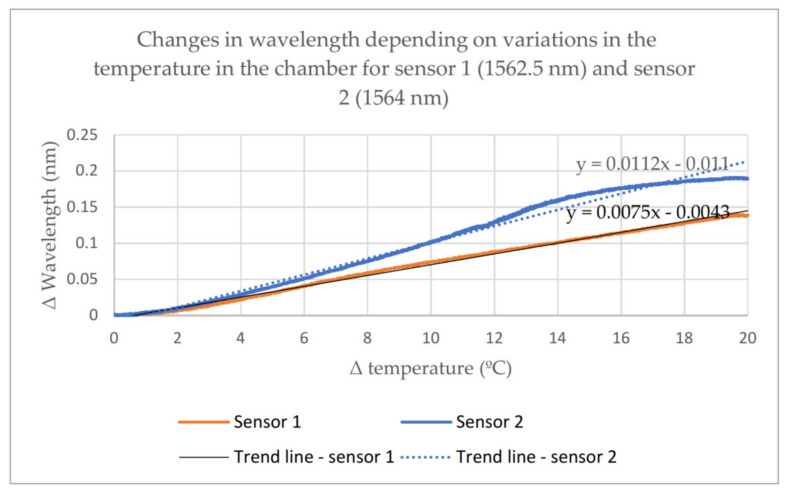
Changes in wavelength depending on temperature changes—sensor 1 and sensor 2.

**Figure 6 materials-14-01207-f006:**
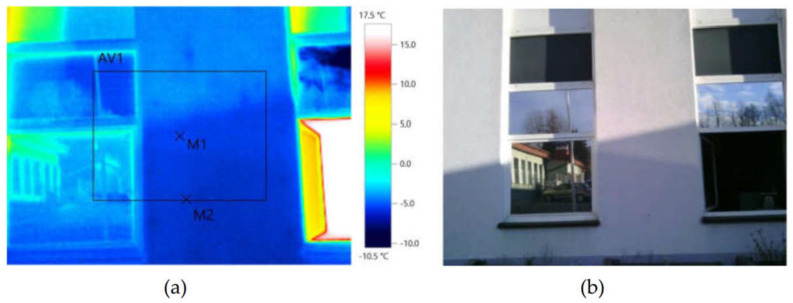
Temperature measurement of the wall from the outside using a thermographic camera (**a**) and the outside view of the wall (**b**).

**Figure 7 materials-14-01207-f007:**
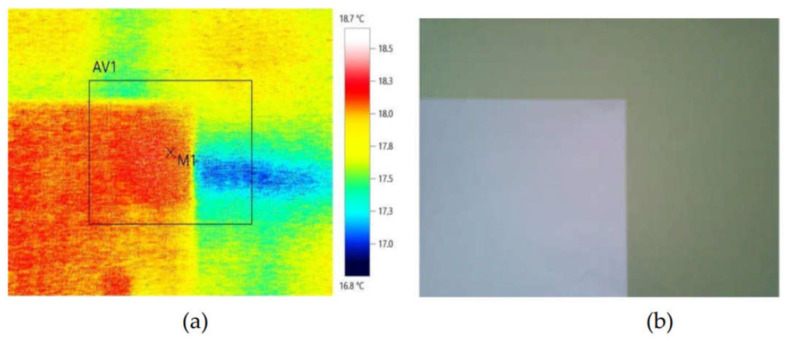
Temperature measurement of the wall from the inside using a thermographic camera (**a**) and the view of the wall and the white sheet of paper (**b**).

**Figure 8 materials-14-01207-f008:**
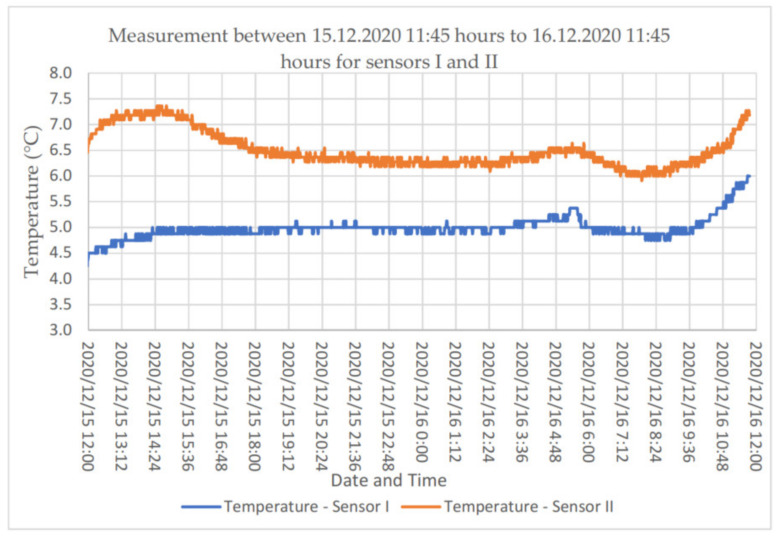
The 24-h temperature distribution for FBG sensors placed in the building envelope.

**Figure 9 materials-14-01207-f009:**
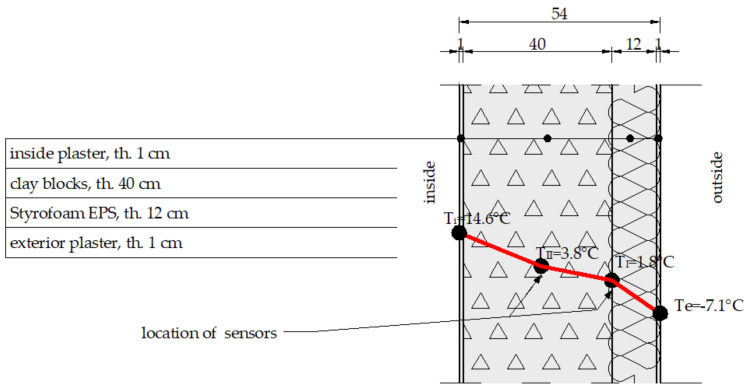
Temperature distribution in the building envelope during the thermographic camera measurement.

**Figure 10 materials-14-01207-f010:**
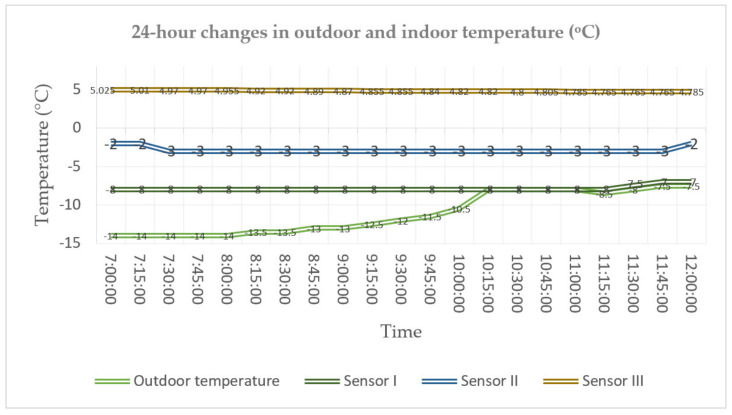
The 24-h distribution of outdoor temperature and temperature inside the building envelope.

**Figure 11 materials-14-01207-f011:**
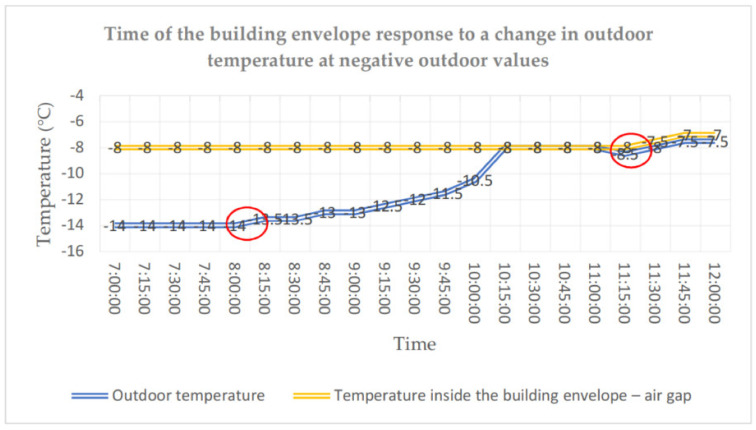
Time of the building envelope response to a change in outdoor temperature at negative outdoor values.

**Figure 12 materials-14-01207-f012:**
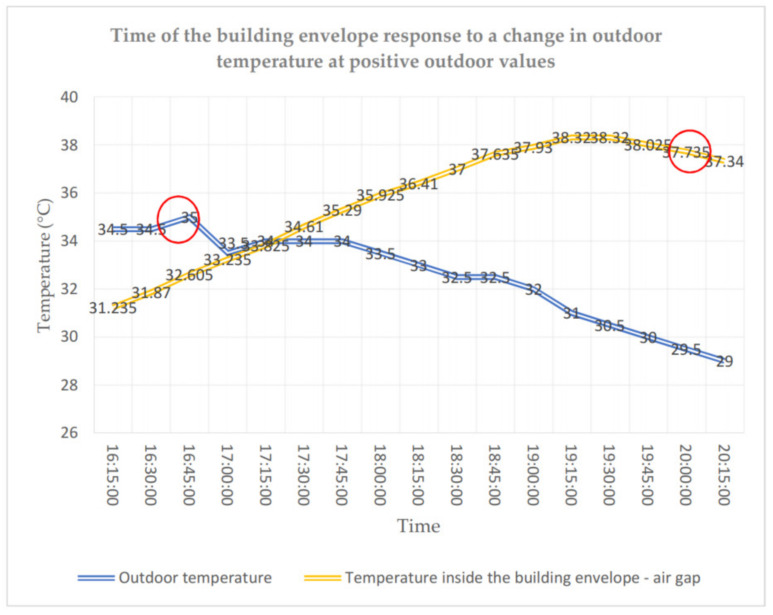
Time of the building envelope response to a change in outdoor temperature at positive outdoor values.

**Figure 13 materials-14-01207-f013:**
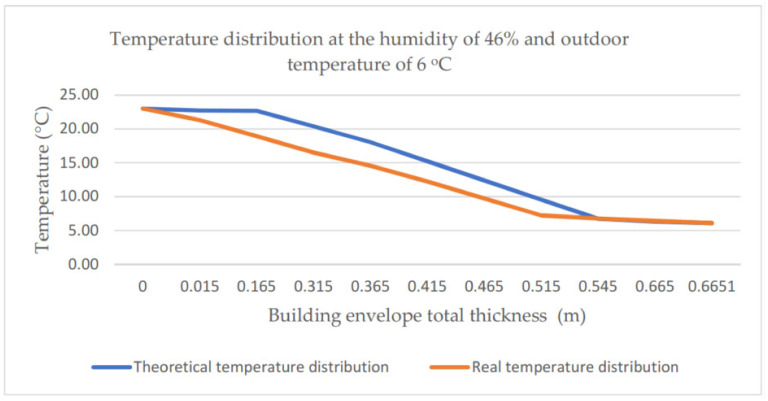
Temperature distribution at the humidity of 46% and outdoor temperature of 6 °C.

**Figure 14 materials-14-01207-f014:**
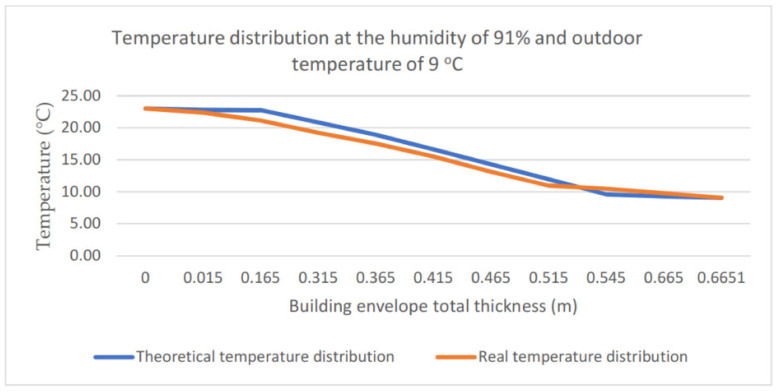
Temperature distribution at the humidity of 91% and outdoor temperature of 9 °C.

**Figure 15 materials-14-01207-f015:**
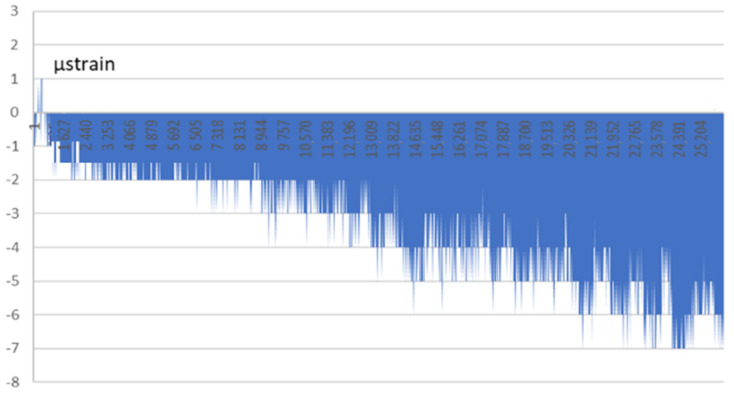
Strain in the building envelope.

**Figure 16 materials-14-01207-f016:**
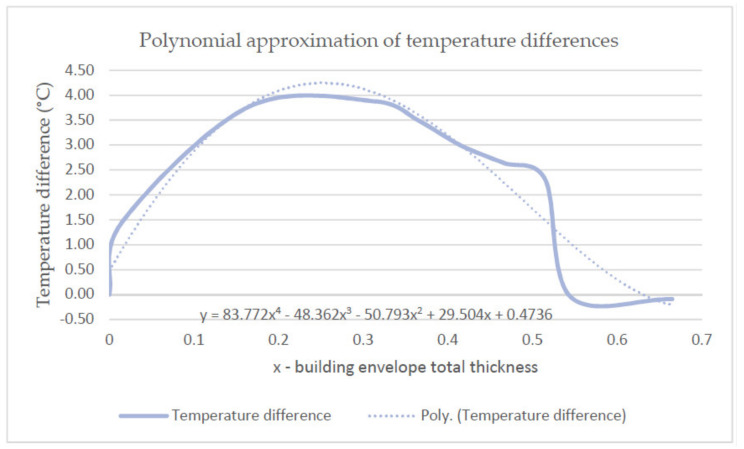
Polynomial approximation of temperature differences.

**Figure 17 materials-14-01207-f017:**
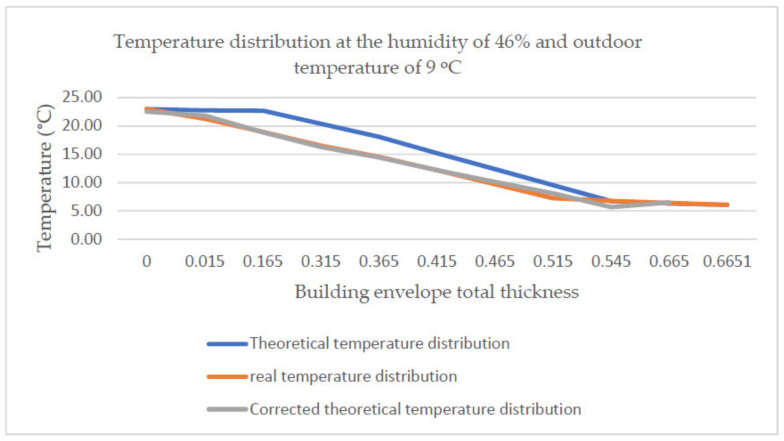
Temperature distribution at the humidity of 46% and outdoor temperature of 9 °C together with the corrected distribution.

**Figure 18 materials-14-01207-f018:**
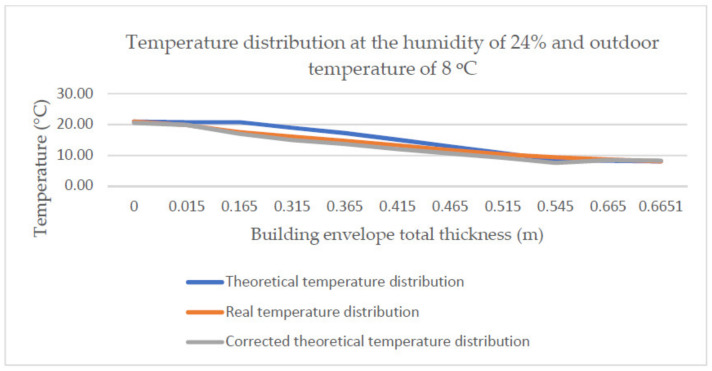
Temperature distribution at the humidity of 24% and outdoor temperature of 8 °C together with the corrected distribution.

**Table 1 materials-14-01207-t001:** Measuring system calibration.

Temperature Accuracy (Guaranteed)	Temperature Accuracy (Typical)	Temperature Precision (Guaranteed)	Temperature Precision (Typical)	Ingress Protection Rating
1 °C	<1 °C	±0.3 °C	±0.2 °C	IP 67

**Table 2 materials-14-01207-t002:** Marking of constant calibration parameters.

Measurand	Description
T (°C)	Temperature
λ_T,act_ (nm)	Actual temp. wavelength
λ_T,ref_ (nm)	Reference temp. wavelength
T_S1_ (°C)	Temperature coefficient 1
T_S2_ (°C)	Temperature coefficient 2
T_S3_ (°C)	Temperature coefficient 3

**Table 3 materials-14-01207-t003:** Calibration coefficients.

T_s1_ (°C)	T_s2_ (°C)	T_s3_ (°C)	λ_Τ,ref_ (nm)
−2,477,681.366	52,768.19076	22.50476755	1,548,514.494
−2,569,888.019	53,819.29229	22.50525754	1,548,477.574
−2,311,1030.106	53,588.17046	22.5027883	1,548,633.785

**Table 4 materials-14-01207-t004:** Temperature values on the surface of the external wall, obtained using the thermographic camera.

Date: 12 January 2021	Time: 8:58:37	Outdoor Temperature: −3.1 °C
Measured quantity		Temperature (°C)
Measuring point 1		−6.5
Measuring point 2		−7.1
Mean value of the surface measurements		−5.9

**Table 5 materials-14-01207-t005:** Linear correlation between the temperature inside the building envelope and the outdoor temperature.

Temperature Sensor in the Envelope	Linear Correlation (*p* < 0.05000 N = 2689) in the Winter Period
	Outdoor Temperature
Sensor I	0.937465
Sensor II	0.843755
Sensor III	0.760560

**Table 6 materials-14-01207-t006:** Polynomial approximation of the temperature distribution.

Theoretical Temperature on the Interface of Layers	Real Temperature	Temperature Difference	Polynomial Approximation of Differences (Correction)	Post-Correction Temperature on the Interface of Layers
23.00	23.00	0.00	0.47	22.53
22.72	21.27	1.45	0.90	21.82
22.68	18.91	3.77	3.80	18.88
20.37	16.50	3.88	4.04	16.33
18.06	14.57	3.50	3.61	14.45
15.22	12.23	2.99	3.00	12.23
12.39	9.74	2.65	2.26	10.12
9.55	7.25	2.30	1.48	8.07
6.71	6.76	−0.05	1.03	5.68
6.33	6.42	−0.09	−0.21	6.54
6.09	6.09	0.00	−0.21	6.29

## Data Availability

The data presented in this study are available on request from the corresponding author.

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
