# Peer review of "Fibre Optic FBG Sensors for Monitoring of the Temperature of the Building Envelope"

_materials, 2021, doi:10.3390/ma14051207_

Round 1

Reviewer 1 Report

I think the readers will appreciate the results of this manuscript.  Generally speaking, the manuscript is well written, the material is judiciously divided and organized and correct from scientific point of view. Some changes are, however, necessary. For these reasons I can recommend the acceptance of this paper after some corrections.

Before that the Editor makes a decision, I suggest that the authors emphasize take into account the following corrections

  1. In Introduction, author described past work, but little comment on the contribution and shortcoming. Author need to provide critical comments.
    2. Please highlight how the work advances or increments the field from the present state of knowledge and provide a clear justification for your work.
    3. In the conclusion, please show how the work advances the field from the present state of knowledge. Please provide a clear justification for your work in this section, and indicate uses and extensions if appropriate. Moreover, you can suggest future developments in the domain.

Author Response

The approach based on special optical FBG temperature sensors to monitor the temperature distribution in a building wall has never been proposed before. Another novelty is the application of optical FBG sensors to measure not only the wall temperature but also the wall strain. The proposed FBG sensors have a very small diameter, which is of special significance because they can be installed in existing walls of already operated buildings. Temperature measurements in external walls of buildings are necessary because based on them it is possible to select the optimal heating system depending on the wall structure and specific climatic conditions of the building location. Moreover, the knowledge of real delays of the temperature distribution in the wall due to changes in the outdoor temperature makes it possible to take adequate action in the HVAC system in advance. Furthermore, the knowledge of the real temperature distribution in the wall enables verification of the theoretical (standard) temperature distribution, introduction of a correction factor, better selection of the heating system parameters and more precise identification of places in the wall where water vapour condensation occurs. The rationale for presenting walls with traditional and optical FBG temperature sensors in the paper was the need to draw attention to a rather high failure frequency of the former. What is also important, previous works do not take up the issues of the life and failure frequency of classical sensors intended for measurements of temperature. Such measurements are carried out on a long-term basis (over the period of many years). A failure of a temperature sensor and the time drift create serious problems in the analysis of results. The operational analysis of classical temperature sensors embedded in the building walls conducted by the authors over the period of 4 years showed their 19% failure rate. In a building that is already operated their replacement is practically impossible, which is a significant problem. The number of sensors installed in the analysed facility was about 3000. The only feasible option in an already operated building is to introduce optical FBG sensors into the wall in place of the damaged classical ones. It should also be noted that in the same period of observation no failure occurred of any of the optical FBG sensors. FBG sensors can be installed in place of damaged traditional sensors owing to their small dimensions, and that was done also in the case of the analysed three-layer wall, which may not have been described precisely enough. FBG sensors are not only equivalent to classical sensors but they also demonstrate a number of advantages that the latter are practically unable to achieve. The system of temperature measurements in the building wall presented in the paper and based on optical FBG sensors ensures unprecedented measuring stability, operational durability and the possibility of creating networks. Moreover, it enables strain and humidity measurements. The significant development of the fibre optic technology, and especially of the design of optical interrogators and fibre optic sensors, will make it possible to introduce into the building structure a “nervous system” that comprehensively and simultaneously monitors temperature, strain, stress or vibration of the building walls. Considering the latest technological progress in the field of optical interrogators and optical FBG sensors, their cost is comparable to classical methods of temperature measurement in the building wall, whereas the computational cost is acceptable for large facilities. The outdoor temperature in both cases of the walls under analysis was obtained from a weather station, while the indoor temperature was measured with a sensor type Testo 605i. The theoretical distribution of temperature is determined in the paper based on analytical formulas and is included in the PN ISO 13788:2013-05 standard.

The core of the paper was effective determination of real temperature distributions in two types of walls (two- and three-layer) of a university building and their comparison with theoretical ones. An analysis of the impact of weather factors was performed next. It was found that changes in air humidity involved differences in temperature distributions in the three-layer wall, while no such difference was observed in the two-layer wall. This can be taken into account in the selection of the heating system parameters and optimization of the system control.

Reviewer 2 Report

According the abstract and the conclusions, the paper proposes a novel way to measure the inner temperature in the external walls of a Building based on FBG sensors. Nevertheless, according the bibliography cited in the paper, this approach has been proposed before. Therefore, which is the paper novelty?

In the introductions authors review several previous works, but they do not explain why is necessary the measurements of the temperature in external walls of a building.

On the other hand there are many aspects of the paper that are not clear. According the experimental description in chapter 2, two types of walls have been studied, but in one case, traditional temperature sensors were used (line 112). Which is the reason for including those measurements if the work deals with the use of FBG temperature sensors? Moreover the experimental procedure used in the 2 layers wall and the 3 layers wall are completely different. The inside and outside temperature is determined by a thermographic camera in one case and in the another a weather station is used. Is in this later case, where non FBG sensor are used when author analyze the influence of other parameters like humidity, which is not related to the core of the work according the title, abstract and conclusions. What authors mean with theoretical temperature distribution? How is it obtained?

Other minor aspects to be clarified or corrected are the following:

Are the FBG sensor used commercially available? If so, can you include the manufacturer information?

The FBG acronym is defined after being used in the text several times. The acronym must be introduced the first time it is used.

In Figure 1, what is λ?

In Figures 1, 2 and 11 units are not always written with an space between the numerical value and the unit as the International System of Units (SI) rules.

Is Table 1 showing the calibration results or the specifications of the manufacturer? I fit shows the results of the calibration process, a detailed description of the experiments carried out to determine the parameters should be provided.

In Table 2, the use of the term sensitivity for TS1, TS2, TS3 is confusing. In fact these parameters are the coefficients of the 2 order polynomial.

In Table 3 the units for λ I assume are [nm] instead of [ºC]. How are these parameters obtained? Was the calibration procedure performed using the climatic chamber?

In line 128 authors mention a strain sensor. Which is this sensor and for what is used?

How is Formula 1 obtained?

Why the sampling rate has been fixed to 1 sample/s. Usually temperature changes are not so fast. Moreover, authors explain later that a program has been developed to get a sample per minute. This program calculates an average of the measurements performed each minute or it just takes 1 of the 60 samples?

When using the climatic chamber, the temperature range is -20 to 40 ºC, but which is the temperature change rate? How many times is the cycle repeated?

Figures 3 and 4 (I suggest to use a single Figure with different labels) show results for sensor 1 and 2. What about sensor 3? At least in Table 3 3 sensors are listed. Those Figures show the change in wavelength due to a change in temperature, but which is the reference temperature?

Why those measurements are presented in the chapter of materials and methods instead of being placed in the chapter devoted to results? The same applies for the thermographic camera?

Is Figure 9 really relevant, since Figure 10 contains the same information

Author Response

   Replies to Reviewer’s/ Reviewers’ comments

The approach based on special optical FBG temperature sensors to monitor the temperature distribution in a building wall has never been proposed before. Another novelty is the application of optical FBG sensors to measure not only the wall temperature but also the wall strain. The proposed FBG sensors have a very small diameter, which is of special significance because they can be installed in existing walls of already operated buildings. Temperature measurements in external walls of buildings are necessary because based on them it is possible to select the optimal heating system depending on the wall structure and specific climatic conditions of the building location. Moreover, the knowledge of real delays of the temperature distribution in the wall due to changes in the outdoor temperature makes it possible to take adequate action in the HVAC system in advance. Furthermore, the knowledge of the real temperature distribution in the wall enables verification of the theoretical (standard) temperature distribution, introduction of a correction factor, better selection of the heating system parameters and more precise identification of places in the wall where water vapour condensation occurs. The rationale for presenting walls with traditional and optical FBG temperature sensors in the paper was the need to draw attention to a rather high failure frequency of the former. What is also important, previous works do not take up the issues of the life and failure frequency of classical sensors intended for measurements of temperature. Such measurements are carried out on a long-term basis (over the period of many years). A failure of a temperature sensor and the time drift create serious problems in the analysis of results. The operational analysis of classical temperature sensors embedded in the building walls conducted by the authors over the period of 4 years showed their 19% failure rate. In a building that is already operated their replacement is practically impossible, which is a significant problem. The number of sensors installed in the analysed facility was about 3000. The only feasible option in an already operated building is to introduce optical FBG sensors into the wall in place of the damaged classical ones. It should also be noted that in the same period of observation no failure occurred of any of the optical FBG sensors. FBG sensors can be installed in place of damaged traditional sensors owing to their small dimensions, and that was done also in the case of the analysed three-layer wall, which may not have been described precisely enough. FBG sensors are not only equivalent to classical sensors but they also demonstrate a number of advantages that the latter are practically unable to achieve. The system of temperature measurements in the building wall presented in the paper and based on optical FBG sensors ensures unprecedented measuring stability, operational durability and the possibility of creating networks. Moreover, it enables strain and humidity measurements. The significant development of the fibre optic technology, and especially of the design of optical interrogators and fibre optic sensors, will make it possible to introduce into the building structure a “nervous system” that comprehensively and simultaneously monitors temperature, strain, stress or vibration of the building walls. Considering the latest technological progress in the field of optical interrogators and optical FBG sensors, their cost is comparable to classical methods of temperature measurement in the building wall, whereas the computational cost is acceptable for large facilities. The outdoor temperature in both cases of the walls under analysis was obtained from a weather station, while the indoor temperature was measured with a sensor type Testo 605i. The theoretical distribution of temperature is determined in the paper based on analytical formulas and is included in the PN ISO 13788:2013-05 standard.

The core of the paper was effective determination of real temperature distributions in two types of walls (two- and three-layer) of a university building and their comparison with theoretical ones. An analysis of the impact of weather factors was performed next. It was found that changes in air humidity involved differences in temperature distributions in the three-layer wall, while no such difference was observed in the two-layer wall. This can be taken into account in the selection of the heating system parameters and optimization of the system control.

Other minor aspects to be clarified or corrected are the following:

Are the FBG sensor used commercially available? If so, can you include the manufacturer information? The sensor is made by the Sylex company.

The FBG acronym is defined after being used in the text several times. The acronym must be introduced the first time it is used.

In Figure 1, what is λ?

In Figures 1, 2 and 11 units are not always written with an space between the numerical value and the unit as the International System of Units (SI) rules.

Relevant corrections have been made.

Is Table 1 showing the calibration results or the specifications of the manufacturer? I fit shows the results of the calibration process, a detailed description of the experiments carried out to determine the parameters should be provided. These are the results of calibration carried out at the manufacturer’s laboratory.

In Table 2, the use of the term sensitivity for TS1, TS2, TS3 is confusing. In fact these parameters are the coefficients of the 2 order polynomial.  The comment has been taken into account and the coefficients have been defined appropriately.

In Table 3 the units for λ I assume are [nm] instead of [ºC]. How are these parameters obtained? Was the calibration procedure performed using the climatic chamber?

This is how the nm unit was corrected. The calibration procedure was carried out in a climatic chamber.

In line 128 authors mention a strain sensor. Which is this sensor and for what is used? Corrected

How is Formula 1 obtained? Formula 1 was determined at the manufacturer’s laboratory by measuring the change in the length of the Bragg grating and the corresponding change in the wavelength due to a change in temperature in the certified climatic chamber.

Why the sampling rate has been fixed to 1 sample/s. Usually temperature changes are not so fast. Moreover, authors explain later that a program has been developed to get a sample per minute. This program calculates an average of the measurements performed each minute or it just takes 1 of the 60 samples?

The lowest sampling frequency of the FBG-800 optical interrogator is 1 HZ.   The program averages the value from 60 samples because the differences between individual measurements are minimal.

When using the climatic chamber, the temperature range is -20 to 40 ºC, but which is the temperature change rate? How many times is the cycle repeated? The cycle was repeated twice (hysteresis and uncertainty can be determined)

Figures 3 and 4 (I suggest to use a single Figure with different labels) show results for sensor 1 and 2. What about sensor 3? At least in Table 3 3 sensors are listed. Those Figures show the change in wavelength due to a change in temperature, but which is the reference temperature?

Sensor 3 concerns the three-layer wall.

It is not 3 sensors but 3 values of constants that are given in Table 3

The rate of temperature changes in the climatic chamber was …..m/s.  The reference temperature was 20oC.

Why those measurements are presented in the chapter of materials and methods instead of being placed in the chapter devoted to results? The same applies for the thermographic camera?

In this section the temperature measurement method based on optical FBG sensors is discussed and preliminary testing is carried out. The outdoor temperature was in each case established by a weather station, whereas the room temperature was measured using a calibrated instrument type… The thermographic camera was an additional tool and enabled evaluation of the experiment boundary conditions. The results obtained from the measurements performed using the camera coincided with the results given by the instrument type TESTO 605i The title of the section has been extended by adding: and preliminary testing.

Is Figure 9 really relevant, since Figure 10 contains the same information

Following the Reviewer’s suggestion, the figure has been removed.

Reviewer 3 Report

This paper proposes to use fibre optic FBG sensors for monitoring the temperature of the building envelope. Overall, the structure of this paper is well organized, and the presentation of this paper is clear. However, there are still some crucial issues that need to be carefully addressed before a possible publication. More specifically,

(1) In the title, “remote” -> ”remotely”?

(2) In the introduction part, a deep literature review should be given, particularly using remote sensing techniques to identify the materials, e.g., “Land surface temperature retrieval from Landsat 8 OLI/TIRS images based on back-propagation neural network. Indoor and Built Environment, 2021,1420326X19882079.” and “An augmented linear mixing model to address spectral variability for hyperspectral unmixing. IEEE Transactions on Image Processing, 2019, 28(4), 1923-1938.”

(3) A proposed workflow should be provided to give the readers a big picture. (4) How about the efficiency (running time and computational cost)?

Author Response

Reply to comments of Reviewer 3:

All comments have been taken into account and relevant corrections have been made in the text.

  1. The title has been changed – “remote” was deleted.
  2. The Reviewer's suggestions have been followed in the Introduction: the literature review has been supplemented with the suggested literature item related to the subject matter of the paper.
  3. The comment related to the method efficiency has been used to supplement the conclusions.

Considering the latest technological progress in the field of optical interrogators, the cost is comparable to classical methods of temperature measurements in the building wall. In fact, it can even be included in the low-cost group, while the computational cost is acceptable for large facilities.

Round 2

Reviewer 2 Report

Authors have clarified the novelty of the paper in the introduction, but they express that the type of sensors used can be also used to measure strain, but this possibility is not presented in this paper. Therefore, this fact must be clarified in the introduction. They have also justified why this type of mesurements are required  and they justify why some mesurements are performed with the FBG sensors and other ones using the resistive ones, but this fact does not appear in the abstract and neither discussed in the results, or in the conclusions.

They have merged or supressed some Figures as suggested, but the structure of the paper has not been changed. This structure make the paper hard to read creating also confusion to the reader.

Answers to the formulatd questions in the previous revision has not been reflected in the paper. Moreover, in the reply letter authors afirm that they have changed the title of section 2, but this change is not reflected in the new paper version.

Some errors still remain or changes have been performed in a wrong way:

In Figure 2 two units are written without a space between the value and the unit.

In table 3, units for the temperature coeficeints should be ºC, not nm

Author Response

Those comments are all valuable  and  very  helpful  for  revising  and  improving  our  paper,  as  well  as the   important   guiding   significance   to   our   researches. We have studied comments carefully and have made correction which we hope meet with approval. Revised portion are marked in red in the paper.

Reviewer 3 Report

No more comments.

Author Response

Those comments are all valuable  and  very  helpful  for  revising  and  improving  our  paper,  as  well  as the   important   guiding   significance   to   our   researches. We have studied comments carefully and have made correction which we hope meet with approval. Revised portion are marked rec.1,2.3 in the paper.

Round 3

Reviewer 2 Report

Although authors have corrected some issues of the paper and clarified some aspects, it is still hard to read. Moreover, new errors have been detected in the new parts:

In line 353 5kN should be written 5 kN

In line 356 100Hz should be written 100 Hz